# Adjustment of Immunosuppressants to Facilitate Anti-COVID-19 Antibody Production after mRNA Vaccination in Liver Transplant Recipients

**DOI:** 10.3390/v15030678

**Published:** 2023-03-04

**Authors:** Wei-Chen Lee, Hao-Chien Hung, Jin-Chiao Lee, Chung-Guei Huang, Po-Wei Huang, Po-Wen Gu, Yu-Chao Wang, Chih-Hsien Cheng, Tsung-Han Wu, Chen-Fang Lee, Ting-Jung Wu, Hong-Shiue Chou, Kun-Ming Chan

**Affiliations:** 1Division of Liver and Transplantation Surgery, Department of General Surgery, Chang-Gung Memorial Hospital, Taoyuan 33357, Taiwan; 2Surgery, College of Medicine, Chang-Gung University, Taoyuan 33357, Taiwan; 3Department of Laboratory Medicine, Chang-Gung Memorial Hospital, Taoyuan 33357, Taiwan; 4Department of Medical Biotechnology and Laboratory Science, Chang-Gung University College of Medicine, Taoyuan 33357, Taiwan

**Keywords:** COVID-19, vaccine, liver transplantation, Moderna mRNA-1273 vaccine, immunosuppressant

## Abstract

Liver transplant recipients are immunocompromised and have low immunogenicity to produce antibodies in anti-COVID-19 vaccination. Whether immunosuppressant adjustment could facilitate anti-COVID-19 antibody production in anti-COVID-19 mRNA vaccination is undetermined. Our patients were informed to temporarily suspend mycophenolate mofetil (MMF) or everolimus (EVR) for 2 weeks during both the 1st and 2nd doses of Moderna mRNA-1273 vaccine. A total of 183 recipients receiving two doses of Moderna mRNA-1273 vaccine were enrolled and grouped into tacrolimus monotherapy (MT, *n* = 41), and dual therapy with non-adjustment (NA, *n* = 23), single suspension (SS, *n* = 19) and double suspension (DS, *n* = 100) of MMF/EVR in two-dose mRNA vaccination. A total of 155 (84.7%) patients had a humoral response to vaccines in this study. The humoral response rates were 60.9%, 89.5%, 91.0% and 80.5% in NA, SS, DS, and MT group patients, respectively (*p* = 0.003). Multivariate analysis showed that favorable factors for humoral response were temporary suspension of MMF/EVR and monotherapy, and unfavorable factors were deceased donor liver transplantation, WBC count < 4000/uL, lymphocyte < 20% and tacrolimus trough level ≥ 6.8 ng/mL. In conclusion, temporary two-week suspension of anti-proliferation immunosuppressants could create a window to facilitate antibody production during anti-COVID-19 mRNA vaccination. This concept may be applied to other vaccinations in liver transplant recipients.

## 1. Introduction

The COVID-19 pandemic is a severe infectious disease spread since 2019. Millions of people died of the disease or disease-related complications [1]. Although the disease spread may be controlled by quarantining infected people, anti-COVID-19 vaccination will be one of the prompt ways to control the disease [2]. Anti-COVID-19 vaccination has been proven to effectively reduce disease severity, hospitalization and death. Currently, mRNA and recombinated viral vector-based vaccines are applied to vaccinate the general population and yield antibody production in 64–98% of the vaccinated people [2]. Recently, the Omicron COVID-19 variant has emerged in a new wave of infection [3] and booster vaccination is urged.

Organ transplant recipients are a special group of patients who need long-term immunosuppressive agents to prevent organ rejection. Although the immunosuppressive agents are minimized in long-term transplant recipients, transplant recipients remain immunocompromised and have the high risk to get COVID-19 infection [4]. In a review article, Schinas et al. have mention that chronic liver disease and liver transplantation are both immune dysfunction patients, and severe COVID-19 infection will further dysregulate liver function which increases mortality in these liver disease patients [5]. Therefore, anti-COVID-19 vaccination is urged for these immune-compromised patients. However, the rates of anti-COVID-19 antibody production after vaccination are low in organ transplant recipients. In kidney transplant recipients, only 6.2% of the patients produced antibody after 1^st^ dose of Pfizer-BioNTech BNT162b2 COVID-19 vaccine or Moderna mRNA-1273 vaccine [6]. In lung transplantation, positive anti-spike protein antibody response rate was only 4% after 1^st^ dose of BNT 162b2 vaccine and 18% after the second dose of vaccine [7]. In liver transplantation, only 47.5% of the recipients produced antibodies after two doses of BNT 162b2 vaccine compared to 100% response rate in healthy control, and the antibody levels were two-fold less than healthy controls [8]. 

Immunosuppressants must play a key role in no or low response to anti-COVID-19 vaccines in organ transplant recipients. High-dose steroids and mycophenolate mofetil (cellcept^®^, MMF) were reported as negative predictors of anti-COVID-19 vaccine responses in liver transplantation [8]. Hence, it is essential to adjust immunosuppressants to facilitate humoral response to anti-COVID-19 vaccines. In this study, we adjusted the immunosuppressive regimen by a temporary 2-week suspension of anti-proliferation immunosuppressants in our long-term stable liver transplant recipients when they received Moderna mRNA-1273 vaccines. Herein, we would like to report the responses to anti-COVID-19 Moderna mRNA-1273 vaccine under such an immunosuppressive regimen adjustment. 

## 2. Materials and Methods

### 2.1. Patients

The liver transplant recipients who were regularly followed up at Chang-Gung Memorial Hospital, LinKou and received two doses of Moderna mRNA-1273 vaccine were included in this study. All the patients received the vaccines between June 2021 and November 2021. All patients did not have the history of COVID-19 viral infection and COVID-19 serology test was not performed before anti-COVID-19 vaccination. All patients signed the consents and prospectively agreed to provide serum for antibody measurement after vaccination. Neutralizing antibody assay was performed between 1 to 2 months after the last vaccination. This study conformed to the ethical guidelines of the 2000 Declaration of Helsinki and was approved by institutional review board of Chang-Gung Memorial Hospital (IRB No. 202102089B0).

### 2.2. Cell Culture and Virus

African green monkey kidney (Vero E6) cells (CRL-1586) were purchased from the American Type Culture Collection (ATCC, Bethesda, MD, USA) and maintained in Dulbecco’s modified Eagle’s medium (DMEM; Gibco, Waltham, MA, USA) containing 10% fetal bovine serum (FBS; Gibco) at 37 °C. Isolated severe acute respiratory syndrome coronavirus 2 (SARS-CoV-2/human/TWN/CGMH-CGU-01/2020) was used in the live virus microneutralization assay.

### 2.3. Wild-Type COVID-19 Neutralizing Antibody Assay

Blood samples were obtained and separated into plasma within 4 h of collection. Anti-live-COVID-19 spike-neutralizing antibody levels were measured as described previously [9]. The live viral culture procedures were conducted in a biosafety level 3 facility under regulations of the Taiwan Center for Disease Control. Briefly, Vero E6 cells were seeded in 96-well plates and incubated overnight. Tested sera were diluted in modified Eagle’s medium (MEM; Thermo Fisher Scientific) at an initial dilution factor of 20, and then further two-fold serial dilutions were performed to a final dilution of 1:5120. COVID-19 virus at 100 TCID50/50 μL (Wuhan wildtype, hCOVID-19/Taiwan/CGMH-CGU-01/2020, GenBank accession MT192759) were mixed with sera in an equal volume and incubated at 37 °C for 1 h before adding to Vero E6 cells. The mixture was incubated at 37 °C for five days and then fixed by 4% formalin for 1 h and stained with 0.1% crystal violet for visualization. The quantitative assay of the COVID-19 50% neutralization titer (NT_50_) was determined by the median tissue culture infectious dose (the dilution of a virus required to infect 50% of given cell culture) and converted into an international unit (IU/mL, WHO Standardized) [10]. An equation for converting geometric mean titers was Y = 1.0334X + 1.0103 where Y was log2 value of IU/mL and X was the log2 value of geometric mean titers. The quantitative assay of neutralization titer (NT_50_) was equal to 9.62 IUml. Thus, results displaying below 9.62 IU/mL were considered to be negative and those equal to or above 9.62 IU/mL were considered as a positive humoral response.

### 2.4. Adjustment of Immunosuppressive Regimen

The immunosuppressive regimens were adjusted during Moderna mRNA-1273 vaccination if anti-metabolite immunosuppressant or the mammalian target of rapamycin (mTOR) inhibitor was included in the immunosuppressive regimens. The patients should have their liver transplantation for more than 6 months and have stable liver function for more than 3 months. The immunosuppressive regimen was not adjusted if the immunosuppressive regimen was monotherapy (MT) with tacrolimus (prograf^®^ or adavgraf^®^). The immunosuppressive regimens were adjusted if the immunosuppressive regimens were tacrolimus plus mycophenolate mofetil (cellcept^®^, MMF) or everolimus (certican^®^, EVR). The patients were informed to suspend MMF or EVR for 2 weeks from the date of vaccine injection in both the 1st and 2nd doses of vaccination. For analysis, the patients were further grouped into non-adjustment (NA), single suspension (SS) and double suspension (DS) of MMF/EVR groups in two-dose vaccination according to the adherence to medical advice.

### 2.5. Clinical Following Up

All the patients were regularly followed up at outpatient clinic (OPD) every 1–2 months. Liver function, renal function and trough levels of tacrolimus were all measured every visit at OPD.

### 2.6. Statistical Analysis

Mean values ± standard deviations and numbers with percentages were used for continuous and categorical factors. Variable comparisons were conducted by Pearson’s chi-square test, independent T-test, or nonparametric method. All continuous factors were bisected by the laboratory reference or the optimal cut-off point decided by the Youden index if lack of credible reference for improving clinical usability. A binary logistic regression model was utilized to evaluate the effect of each parameter on the dichotomous event (positive and negative humoral response). In the univariate analysis, variables with a *p*-value <0.100 were considered as potential risks. We proceeded with these candidates for a multivariate analysis to determine independent risk factors. Area under the receiver operating characteristic curve (AUROC) was used to evaluate the predictive value of predictors on an event. Statistical analyses were calculated by the IBM SPSS^®^ version 24.0 (SPSS Incorporation, Chicago, IL, USA) software, and a significant result was defined when a two-tailed *p*-value was <0.05.

## 3. Results

### 3.1. Patients

A total of 183 liver transplant recipients receiving two doses of Moderna mRNA-1273 vaccine were enrolled. Among the patients, the mean age was 65.7 ± 7.7 years old and living donor liver transplantation was more prevalent (*n* = 141, 77.0%). The mean time from liver transplantation to completion of two-dose vaccination was 108.8 ± 59.9 (range: 8.7 to 283.7) months. The most common underlining disease of liver transplantation was viral hepatitis (*n* = 151, 82.5%). Regarding co-morbidities, most of the recipients had fair kidney function and the mean eGFR was 57.4 ± 30.2 mL/min/1.73 m^2^. However, 17 (9.3%) recipients needed enduring hemodialysis (Table 1).

### 3.2. Immunosuppressive Regimens

Tacrolimus was the backbone of our post-transplant immunosuppressive regimen. Among 183 liver transplant recipients, 41 (22.4%) patients had tacrolimus monotherapy (MT group) and 142 (77.6%) patients had tacrolimus with MMF (*n* = 121) /EVR (*n* = 21). The daily dose of MMF was 500 to 1000 mg, and EVR was 0.5–2 mg. Only 5 of 142 patients had triple therapy with steroid (prednisolone, 5 mg/day). Among the 142 patients with dual or triple therapy, 100 patients had adjustment of immunosuppressive regimens by 2-week suspension of MMF or EVR in both the first and second doses of mRNA vaccine (DS group); 19 patients had 2-week suspension of MMF or EVR in either the first or second dose of mRNA vaccine (SS group), and 23 patients did not have immunosuppressive regimen adjustment (NA group).

### 3.3. Humoral Responses to Moderna mRNA Vaccine

A total of 155 (84.7%) patients had a humoral response to vaccines with the COVID-19NT_50_ ≥ 9.62 IU/mL after two doses of Moderna mRNA-1273 vaccine. For the patients with positive neutralizing antibodies, they had a higher percentage of living donor liver transplantation (*p* = 0.026), longer time after liver transplantation (*p* = 0.023), higher lymphocyte count and lower neutrophil-to-lymphocyte ratio (*p* = 0.027) than the patients with a negative humoral response (see Table 1). The patients with temporary suspension of MMF/EVR had a higher humoral response rate to the Moderna mRNA-1273 vaccine. The humoral response rates were 60.9%, 89.5%, 91.0%, and 80.5% in the NA, SS, DS, and MT group patients, respectively (*p* = 0.003). Age, gender, interval between the two vaccinations, renal function and history of hepatocellular carcinoma were all not significant.

### 3.4. Antibody Production and Liver Function in Temporary Suspension of MMF/EVR

Regarding the influence of immunosuppressant settings on the concentration of generated NT_50_, the values of COVID-19 NT_50_ in different group patients were obtained. The median (interquartile) antibody value of post-vaccination serum NT_50_ were 279.3 (50.61–465.04) IU/mL in DS group patients, which was higher than 82.93 (22.63–229.37) IU/mL in SS group patients (*p* = 0.029) and 25.09 (9.62–455.27) IU/mL in NA group patients (*p* = 0.004), and also had a tendency higher than 145.61 (15.02–455.27) IU/mL in MT group patients (*p* = 0.052) (Figure 1a). Regarding graft safety of suspending immunosuppressive agents, the liver function tests were performed before and after the vaccinations for comparison in DS group patients. The median (interquartile, range) of aspartate aminotransferase (AST) was 20 (16–30.8, 9–316) U/L after vaccinations compared to 19.5 (16–27.8, 10–78) before vaccinations (Figure 1b, *p* = 0.200). The median (interquartile, range) of alanine aminotransferase (ALT) was 20.5 (12–34.8, 6–581) U/L after vaccinations compared to 19.5 (13–30, 2–141) before vaccinations (Figure 1b, *p* = 0.089). The patient with hundreds of units of AST and ALT was proved to have a biliary tract infection simultaneously and resumed to normal liver function after antibiotic treatment. Hence, temporary 2-week suspension of MMF or EVR during vaccination was safe and beneficial in humoral responses to vaccines.

### 3.5. Significance of Tacrolimus and Steroid

As tacrolimus was the backbone of our immunosuppressive regimen, it was concerned whether the trough level of tacrolimus influenced the humoral response to Moderna mRNA-1273 vaccine. The analytic results showed that tacrolimus trough level ≥6.8 ng/mL resulted in lower antibody production after one or two doses of Moderna vaccine. The median (interquartile) COVID-19 NT_50_ was 42.25 (12.01–283.02) U/L in DS group patients with tacrolimus trough level ≥6.8 ng/mL, compared to 309.74 (92.76–465.04) U/L in the patients with tacrolimus trough level <6.8 ng/mL (Figure 2, *p* = 0.003). The result was similar in SS group patients. The median (interquartile) COVID-19 NT_50_ was 22.63 (9.62–22.63) U/L in the patients with tacrolimus trough level ≥6.8 ng/mL, compared to 125.43 (33.74–229.37) U/L in the patients with tacrolimus trough level <6.8 ng/mL (Figure 2, *p* = 0.049). For NA and MT group patients, tacrolimus levels did not show the influence on the production of COVID-19 NT_50_. In the study, only five patients had a low dose of prednisolone to maintain immunosuppression. Among the five patients, only two (40%) of them had a humoral response to Moderna mRNA-1273 vaccines.

### 3.6. Independent Predictors Associated with Negative Humoral Response

In a univariate analysis, potential factors associated with negative humoral response were deceased donor liver transplantation, steroid maintenance, time from liver transplantation to vaccination <120 months, AST ≥34 U/L, ALT ≥36 U/L, WBC count < 4000/uL, lymphocyte < 20%, neutrophil ≥ 74%, NLR ≥ 2.25, no adjustment of immunosuppressants and tacrolimus trough level ≥6.8 ng/mL. When a multivariate analysis was conducted for these potential factors, the results revealed that the independent and unfavorable factors for humoral response were deceased donor liver transplantation (odds ratio (OR): 2.87, *p* = 0.038), WBC count < 4000/μL (OR: 3.96, *p* = 0.017), lymphocyte < 20% (OR: 3.38, *p* = 0.018) and tacrolimus trough level ≥6.8 ng/mL (OR: 3.00, *p* = 0.044). The favorable factors for humoral response were temporary suspension of MMF/EVR (OR: 10.0, *p* = 0.021 for single suspension; OR: 7.69, *p* = 0.001 for double suspension) and monotherapy (OR: 4.54, *p* = 0.026). (Table 2) By combination of independent risk factors (deceased donor liver transplantation, leukocytopenia, lymphocytopenia, non-adjustment of immunosuppressants, and tacrolimus trough level ≥6.8 ng/mL), negative humoral response to Moderna mRNA-1273 vaccinations could be predicted (Figure 3. AUROC: 0.809, 95% CI: 0.708–0.910, *p* < 0.001).

## 4. Discussion

Organ transplant recipients need particular attention during the COVID-19 pandemic. Organ transplant recipients are a group of immunocompromised patients with long-term immunosuppressants to prevent organ rejection. They can easily get critical illness in this COVID-19 pandemic [11,12]. Anti-COVID-19 vaccination is a way to protect transplant recipients from critical illness. However, their neutralized antibody response to mRNA vaccine is lower than that of the healthy control [13,14]. In liver transplantation, Thuluvath et al. reported that 61.3% of liver transplant recipients responded poorly to produce antibodies after two doses of anti-COVID-19 vaccine, compared to 24% of cirrhotic or chronic hepatitis patients [15]. Rabinowich et al. also reported that only 47.5% of liver transplant recipients produced antibody after two doses of Pfizer-BioNTech BNT162b2 COVID-19 vaccine, compared to 100% of the healthy control [8]. Even antibody production responded to the anti-COVID-19 vaccine, the antibody levels were much lower in liver transplant recipients than in the healthy control [8,16,17]. Clearly, immunogenicity in liver transplant recipients was low and antibody response to the anti-COVID-19 vaccine was limited. Immunosuppressive agents applied in transplant recipients must be the key players contributing to low immunogenicity and low antibody response rate. However, immunosuppressants are not recommended to be adjusted during vaccination for fear of rejection [18,19]. Rationally, immunosuppressants were needed to be adjusted to enhance antibody response to vaccines.

Temporary suspension of anti-metabolite or mTOR immunosuppressants created a window to produce anti-COVID-19 antibodies effectively when an mRNA vaccine was administered. The eligible liver transplant recipients were informed to suspend MMF or EVR for two weeks during both the first and second vaccinations if MMF or EVR was one of the immunosuppressants in their immunosuppressive regimens. The adequate antibody production was referred to our wild-type COVID-19 neutralization assay with 50% of cells free from infection, which was ≥9.62 IU/mL [10]. According to the multivariate analysis, suspension of MMF/EVR for 2 weeks was the facilitating factor to produce adequate anti-viral antibodies. A total of 91.0% of the patients who held MMF/EVR for 2 weeks in both the first and second doses of vaccine produced anti-viral neutralizing antibody. The percentage was significantly decreased to 60.9% if MMF/EVR was continued when vaccines were injected. In the literature, only 38.7–75% of the patients could produce adequate antibody after anti-COVID-19 vaccine injections without mention of immnunosuppressive regimen adjustment [8,15,16,17]. MMF inhibits de novo purine synthesis by selectively inhibiting inositol-monophosphate-dehydrogenase and thereby suppresses the proliferation of T- and B-lymphocytes [20]. mTOR affects broad aspects of cellular function such as metabolism, growth, survival, etc. EVR blocks cell cycle progress at the G1 to S phase and inhibits T-lymphocytes proliferation [21]. Primary humoral immune response occurs in 1–2 weeks after first exposure to antigens. Hence, adjustment of immunosuppressive regimen by temporal suspension of anti-proliferation immunosuppressants for 2 weeks in each vaccine injection would be the key to produce adequate anti-viral antibody.

Two-week suspension of anti-proliferation immunosuppressants was safe for long-term liver transplant recipients. Two-week suspension was designed because antibody production by activated B-cells through T-cell help happened between 1–2 weeks after viral infection or anti-viral vaccination [22]. In this study, 77.6% of the patients had immunosuppressive therapy with tacrolimus and MMF or EVR. Among these patients, 70.4% of the patients adhered to medical advice and suspended MMF or EVR for 2 weeks during both doses of Moderna mRNA-1273 vaccine. The liver function, AST and ALT did not change significantly prior to and after the 2-week suspension of MMF/EVR. Rejection attack was the greatest concern when anti-proliferation immunosuppressants were suspended. Daily dosage of immunosuppressants may have a surplus for transplant recipients and temporary suspension of anti-proliferation immunosuppressants is tolerable. In this study, temporary suspension of anti-proliferation immunosuppressants for 2 weeks was safe for liver transplant recipients and facilitated the production of anti-viral antibodies.

Concentration of tacrolimus in the serum also influenced antibody production after Moderna mRNA-1273 vaccination. The patients with trough level of tacrolimus above 6.8 ng/mL had a higher incidence of non-response to Moderna mRNA-1273 vaccination than the patients with trough tacrolimus level <6.8 ng/mL. Cholankeril et al. also mentioned that tacrolimus trough level in the patients with negative antibody response was 6.6 ± 2.2 ng/mL compared to 5.4 ± 2.0 ng/mL in the patients with positive antibody response [23]. Among the patients who temporarily suspended MMF/EVR for 2 weeks and responded to product anti-viral antibody, the levels of neutralizing antibody were significantly decreased if trough level of tacrolimus was >6.8 ng/mL. Tacrolimus is a calcineurin inhibitor and majorly suppresses T-cells. As we know, antibody production is from activated B-cells or plasma cells, which need T-cell help [24]. High level of tacrolimus may suppress T-cells extensively and further influence B-cell activation and antibody production. It is better to keep the trough level of tacrolimus below 6.8 ng/mL to facilitate antibody production during Moderna mRNA-1273 vaccination.

Steroids is a significant immunosuppressant to prohibit antibody production after Moderna mRNA-1273 vaccination. In our immunosuppressive regimen, steroids will be stopped within 3 months after liver transplantation except if the patients’ transplanted livers are for autoimmune or autoimmune-similar diseases. In this study, only five patients have a low dose of steroids. Among these five patients, only two (40%) patients responded to the Moderna mRNA-1273 vaccination. This was significantly lower than other patients without steroids. This result was similar to the report that a high dose of steroids was one of significant factor to negatively affect antibody production after anti-COVID-19 vaccination [8].

White blood cell count and lymphocyte percentage in white blood cell count were significant factors related to antibody production after Moderna mRNA-1273 vaccination in this study. The patients with white blood cell count ≥4000/uL had a 2.55-fold higher chance of producing antibody than the patients with white blood cell count <4000/uL. This result was similar to that the patients with white blood cell count <4000/uL was the risk factor of negative response to antibody production, reported by Cholankeril et al. [23]. Lymphocyte count itself was also important in antibody response after Moderna mRNA-1273 vaccination. The patients with lymphocyte count >20% in white blood cell had 3.07-fold higher to produce antibody than the patients with lymphocyte count ≤20% in white blood cell. Anti-viral antibody was produced by activated B-cells through T-cell help. T-cells and B-cells are both included in peripheral lymphocyte count. The patients with a high white blood cell and high lymphocyte count had a high ability to produce antibody after Moderna mRNA-1273 vaccination.

This study showed that adjustment of MMF/EVR facilitated a humoral response to anti-viral vaccination. This concept can be applied to other vaccines such as the influenza vaccine. However, further studies or clinical trials are essential to prove its efficacy. All the patients in this study were liver transplant recipients. As the liver is recognized as the organ with less incidence of acute cell-mediated rejection, whether the safety of this immunosuppressant adjustment can be translated into other solid organ transplantation is unknown and needs to be studied.

There were limitations in this study. We did not perform cellular response after Moderna mRNA-1273 vaccination. We also did not measure specific anti-COVID-19 antibody profiles including anti-RBD and anti-NCAP antibodies to rule out previous contact with COVID-19 because there were only a few people infected at that time in Taiwan. This study was not a randomized clinical trial. Further studies are needed to verify this concept of immunosuppressant adjustment for liver transplantation. Currently, there are other clinical trials registered (e.g., NCT05490342), and we may wait for the results.

## 5. Conclusions

Anti-COVID-19 mRNA vaccination is a prompt way to protect transplant recipients from COVID-19 infection. Long-term immunosuppressants results in low immunogenicity in transplant recipients and cause low humoral response to anti-viral vaccination. Two-week suspension of anti-proliferation immunosuppressants can create a widow to facilitate antibody production during anti-COVID-19 mRNA vaccination.

## Figures and Tables

**Figure 1 viruses-15-00678-f001:**
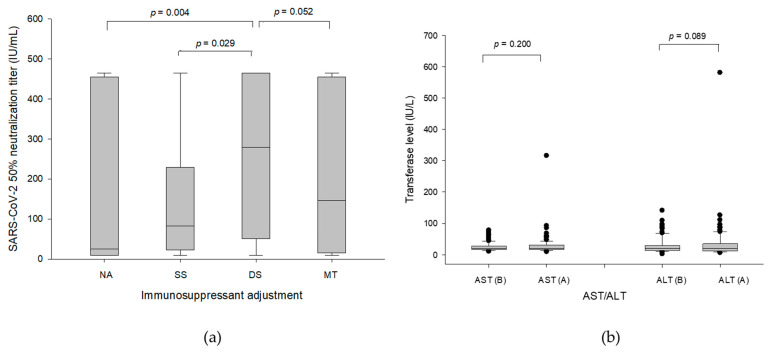
(**a**) The values of COVID-19 NT_50_ in different groups of patients. The median (interquartile) antibody value of post-vaccination serum NT_50_ were 279.3 (50.61–465.04) IU/mL in the DS group of patients, which was higher than 82.93 (22.63–229.37) IU/mL in SS group of patients (*p* = 0.029) and 25.09 (9.62–455.27) IU/mL in NA group patients (*p* = 0.004), and also had a tendency higher than 145.61 (15.02–455.27) IU/mL in MT group patients (*p* = 0.052). (**b**) The liver function tests before and after the vaccinations in DS group patients. The median (interquartile, range) of AST was 20 (16–30.8, 9–316) U/L after vaccinations compared to 19.5 (16–27.8, 10–78) before vaccinations (*p* = 0.200). The median (interquartile, range) of ALT was 20.5 (12–34.8, 6–581) U/L after vaccinations compared to 19.5 (13–30, 2–141) before vaccinations (*p* = 0.089). (B, before vaccination; A, after vaccination).

**Figure 2 viruses-15-00678-f002:**
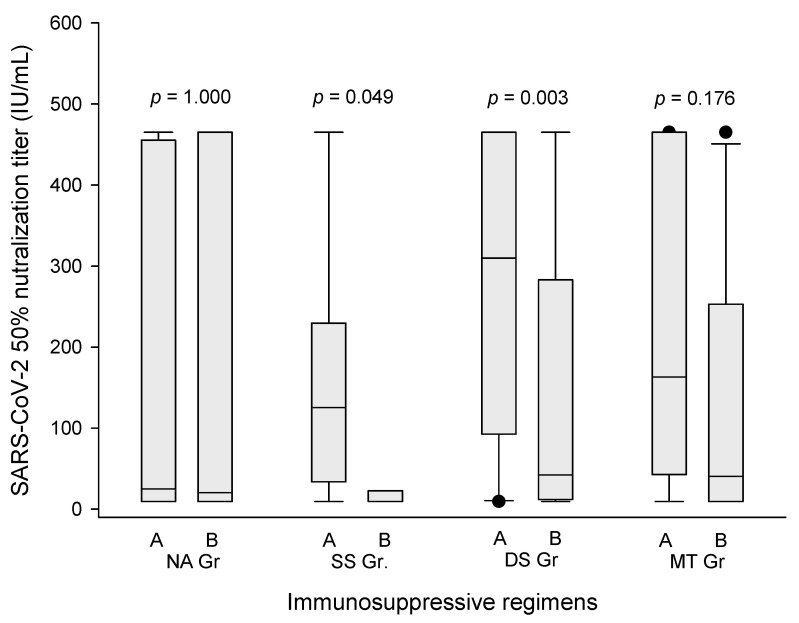
The COVID-19 NT_50_ level according to trough concentration of tacrolimus. The median (interquartile) COVID-19 NT_50_ was 42.25 (12.01–283.02) U/L in DS group patients with tacrolimus trough level ≥6.8 ng/mL (A), compared to 309.74 (92.76–465.04) U/L in the patients with tacrolimus trough level < 6.8 ng/mL (B) (*p* = 0.003). The median (interquartile) COVID-19 NT_50_ in SS group patients was 22.63 (9.62–22.63) U/L in the patients with tacrolimus trough level ≥6.8 ng/mL (A), compared to 125.43 (33.74–229.37) U/L in the patients with tacrolimus trough level < 6.8 ng/mL (B) (*p* = 0.049). For NA and MT group patients, tacrolimus levels did not show the influence on the production of COVID-19 NT_50_.

**Figure 3 viruses-15-00678-f003:**
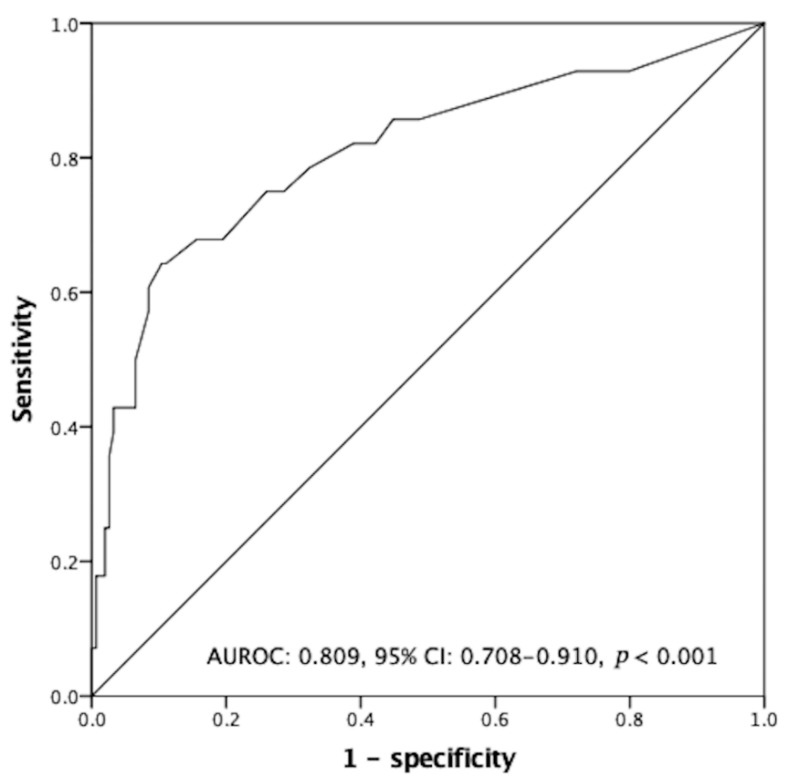
The receiver operating characteristic curve for negative antibody production. By a combination of identified independent risks (deceased donor liver transplantation, leukocytopenia, lymphocytopenia, non-adjustment of immunosuppressants, and tacrolimus trough level ≥6.8 ng/mL) in predicting the development of negative humoral response after two-dose Moderna mRNA-1273 vaccinations (area under ROC: 0.809, 95% CI: 0.708–0.910, *p* < 0.001).

**Table 1 viruses-15-00678-t001:** Comparison of patients’ characteristics according to negative or positive humoral response to two-dose Moderna mRNA-1273 vaccine.

Parameters	All Recipients (*n* = 183)	Negative (*n* = 28)	Positive (*n* = 155)	*p*-Value
Age	65.7 ± 7.7	63.9 ± 7.9	66.2 ± 7.4	0.144
Age, ≥65-year-old	117 (63.9)	15 (53.6)	102 (65.8)	0.215
Gender, male	140 (76.5)	19 (67.9)	121 (78.1)	0.121
Type of transplantation, LDLT	141 (77.0)	17 (60.7)	124 (80.0)	0.026
Maintenance of mTOR inhibitor	21 (11.5)	2 (7.1)	19 (12.3)	0.434
Maintenance of MMF	121 (66.1)	18 (64.3)	103 (66.5)	0.824
Maintenance of corticosteroid	5 (2.7)	3 (10.7)	2 (1.3)	0.005
Time from LT to vaccination, months	108.8 ± 59.9	87.4 ± 54.6	115.5 ± 60.3	0.023
Time from LT to vaccination, ≥120 months	77 (42.1)	6 (21.4)	71 (45.8)	0.016
Interval between two vaccinations, days	83.9 ± 21.2	76.2 ± 25.3	84.8 ± 20.3	0.097
Interval between two vaccinations, ≥28 days	180 (98.4)	28 (100.0)	152 (98.1)	0.458
Immunosuppressant settings				0.003
No adjustment	23 (12.6)	9 (32.1)	14 (9.0)	
Single suspension of MMF/EVR	19 (10.4)	2 (7.1)	17 (11.0)	
Double suspension of MMF/EVR	100 (54.6)	9 (32.1)	91 (58.7)	
Monotherapy (tacrolimus)	41 (22.4)	8 (28.6)	33 (21.9)	
Hx of hepatocellular carcinoma	80 (43.7)	10 (35.7)	70 (45.2)	0.354
Viral hepatitis				0.371
NBNC	32 (17.5)	5 (17.9)	27 (17.4)	
HBV	111 (60.7)	15 (53.6)	96 (61.9)	
HCV	34 (18.6)	8 (28.6)	26 (16.8)	
Co-infection of HBV and HCV	6 (3.3)	0 (0.0)	6 (3.9)	
Hx of alcoholism	22 (12)	6 (21.4)	16 (10.3)	0.096
Hx of autoimmune liver disease	7 (3.8)	1 (3.6)	6 (3.9)	0.939
Hx of any autoimmune disease	10 (5.5)	1 (3.6)	9 (5.8)	0.632
Hx of solid tumor malignancy	87 (47.5)	12 (42.9)	75 (48.4)	0.590
Creatinine, mg/dL	1.8 ± 1.9	2.1 ± 2.4	1.8 ± 1.8	0.410
Creatinine, ≥1.2 mg/dL	84 (45.9)	14 (50.0)	70 (45.2)	0.636
eGFR, mL/min/1.73 m^2^	57.4 ± 30.2	48.9 ± 24.3	59.3 ± 31.1	0.097
eGFP, ≥60 mL/min/1.73 m^2^	84 (45.9)	9 (32.1)	75(48.4)	0.112
Hemodialysis	17 (9.3)	3 (10.7)	14 (9.0)	0.778
Total bilirubin, mg/dL	0.86 ± 0.56	0.85 ± 0.42	0.86 ± 0.61	0.857
Total bilirubin, ≥1.2 mg/dL	24 (13.1)	4 (14.3)	20 (12.9)	0.842
AST, U/L	25.2 ± 16.7	28.2 ± 16.7	23.8 ± 15.3	0.170
AST, ≥34 U/L	24 (13.1)	7 (25.0)	17 (11.0)	0.043
ALT, U/L	28.8 ± 25.8	35.9 ± 32.0	27.1 ± 24.2	0.093
ALT, ≥36 U/L	36 (19.8)	9 (32.7)	27 (17.5)	0.074
Albumin, g/dL	4.3 ± 0.3	4.4 ± 0.3	4.3 ± 0.3	0.498
Albumin, <3.5 g/dL	3 (1.6)	1 (3.6)	2 (1.3)	0.382
WBC count, /uL	5800.1 ± 1682.3	5435.7 ± 2298.8	5869.0 ± 1520.1	0.344
WBC count, <4000/uL	29 (15.8)	8 (28.6)	21 (13.5)	0.045
Neutrophil, %	62.5 ± 9.3	66.0 ± 9.5	62.1 ± 9.2	0.041
Neutrophil, ≥74%	20 (10.9)	7 (25.0)	13 (8.4)	0.010
Lymphocyte, %	27.2 ± 8.8	24.0 ± 9.0	27.9 ± 8.5	0.029
Lymphocyte, <20%	38 (20.8)	11 (39.3)	27 (17.4)	0.009
NLR	2.7 ± 1.7	3.2 ± 1.4	2.6 ± 1.3	0.027
NLR ≥2.25	102 (55.7)	22 (78.6)	80 (51.6)	0.008
Platelet count <100 × 10^9^/L	173.0 ± 55.9	166.8 ± 77.0	174.9 ± 50.3	0.475
Platelet count, <100 × 10^9^/L	13 (7.1)	2 (7.1)	11 (7.1)	0.993
FK506 trough level, ng/mL	5.51 ± 4.41	5.5 ± 3.4	5.1 ± 3.6	0.534
FK506 trough level, ≥6.8 ng/mL	31 (16.9)	9 (32.1)	21 (14.2)	0.020

LT, liver transplantation; IU, international unit; LDLT, living donor liver transplantation; mTOR, mammalian target of rapamycin; MMF, mycophenolate mofetil; NBNC, non-hepatitis B and non-hepatitis C, HBV, hepatitis B virus; HCV, hepatitis C virus; Hx, history; eGFR, estimated glomerular filtration rate; AST, aspartate transaminase; ALT, alanine transaminase; WBC, white blood cell; NLR, neutrophil-to-lymphocyte ratio.

**Table 2 viruses-15-00678-t002:** Uni-/multivariate analyses in predicting negative humoral response after two-dose Moderna mRNA-1273 vaccines in liver transplant recipients.

Parameters (Reference)	Univariate			Multivariate		
	OR	95% CI	*p*-Value	OR	95% CI	*p*-Value
Type of LT, DDLT (LDLT)	2.59	1.10–6.08	0.029	2.87	1.06–7.81	0.038
Maintenance of steroid—yes (no)	9.18	1.46–57.71	0.018			
Triple immunosuppressants—yes (no)	5.89	0.79 –43.64	0.083			
Time from LT to vaccination <120 (≥120) months	3.10	1.19–8.07	0.020	2.97	0.99–8.93	0.053
AST ≥ 34 (<34) U/L	2.71	1.01–7.30	0.049			
ALT ≥ 36 (<36) U/L	2.23	0.91–5.45	0.079			
WBC count < 4000 (≥4000)/uL	2.55	0.99–6.54	0.051	3.96	1.28–12.20	0.017
Lymphocyte < 20 (≥20) %	3.07	1.29–7.28	0.011	3.38	1.24–9.24	0.018
Neutrophil ≥ 74 (<74) %	3.64	1.30–10.17	0.014			
NLR ≥ 2.25 (<2.25)	3.44	1.32–8.94	0.011			
Temporary suspension of immunosuppressant						
Single suspension (NA)	0.18	0.34–0.99	0.049	0.10	0.02–0.71	0.021
Double suspension (NA)	0.15	0.05–0.45	0.001	0.13	0.04–0.45	0.001
Monotherapy (NA)	0.38	0.12–1.18	0.093	0.22	0.06–0.84	0.026
FK506 trough level ≥ 6.8 (<6.8) ng/mL	2.84	1.14–7.08	0.025	3.00	1.03–8.72	0.044

LT, liver transplantation; AST, Aspartate Transaminase; ALT, Alanine transaminase; WBC, white blood cell; NLR, neutrophil-to-lymphocyte ratio; NA, no adjustment. Only significant results (*p*-value < 0.100) were shown in this table and entering the multivariate analysis.

## Data Availability

The authors confirm that the data supporting the findings of this study are available within the article.

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
