# Peer review of "Adjustment of Immunosuppressants to Facilitate Anti-COVID-19 Antibody Production after mRNA Vaccination in Liver Transplant Recipients"

_viruses, 2023, doi:10.3390/v15030678_

Round 1
Reviewer 1 Report
This is an interesting study relating to temporary suspension of immunosuppression in liver transplant recipients so as to achieve better efficacy following COVID-19 vaccination. Overall it is an interesting report I however have some comments
1. The authors should take advantage relative info and cite recently published article by Schinas et al Viruses 2022, 14(12), 2778; https://doi.org/10.3390/v14122778 for their introduction/discussion sections
2. Why the timeline of 2w suspension and not for example 3 was chosen at the first place ? The lack of controls i.e without regimen suspension is a serious limitation in assessing results. Pts might elicit some degree of responses anyhow.
3. A limitation section and a comment on future trials e.g https://clinicaltrials.gov/ct2/show/NCT05490342 should be added in this study
4. Important similar study is missing to discuss similarities and differences (Vaccines, 2022, 29;10(11):1827)
5 A comment on cellular response should be added
6. I would also like to see for better comprehension authors describe mechanisms these immunosuppressants have on humoral reponse
7. LTR also receive corticosteroids, please discuss
Author Response
Dear Reviewer:
We are pleased that we have the opportunity to revise our manuscript. We thank your comments and suggestions. Point-to-point responses are described as following:
- The authors should take advantage relative info and cite recently published article by Schinas et al Viruses2022, 14(12), 2778; https://doi.org/10.3390/v14122778 for their introduction/discussion sections
Reply: We add some description in Introduction section and cite Dr. Schinas’ paper.
- Why the timeline of 2w suspension and not for example 3 was chosen at the first place ? The lack of controls i.e without regimen suspension is a serious limitation in assessing results. Pts might elicit some degree of responses anyhow.
Reply: We add discussion in discussion section “Primary humoral immune response occurs in 1-2 weeks after first exposure to antigens.” and the limitation paragraph in Discussion section.
- A limitation section and a comment on future trials e.g https://clinicaltrials.gov/ct2/show/NCT05490342 should be added in this study
Reply: A limitation paragraph was added in Discussion section.
- A comment on cellular response should be added.
Reply: We did not perform cellular response in this study. This is a limitation of our study.
- I would also like to see for better comprehension authors describe mechanisms these immunosuppressants have on humoral reponse
Reply: The mechanisms of the immunosuppressants were added in Discussion section: “ MMF inhibits de novo purine synthesis by selectively inhibiting inositol-monophosphate-dehydrogenase and thereby suppresses the proliferation of T- and B-lymphocytes.mTOR affects broad aspects of cellular function such as metabolism, growth, survival, etc. EVR blocks cell cycle progress at the G1 to S phase and inhibit T-lymphocytes proliferation歐”
- LTR also receive corticosteroids, please discus
Reply: we add a paragraph to discuss steroid in Discussion Section.
Thank you for your comments and suggestions again. Hopefully, our manuscript can be accepted.
Sincerely yours,
Wei-Chen Lee
Reviewer 2 Report
This study is a prospective, single-center study designed to promote antibody acquisition by Moderna mRNA1273 vaccination to prevent COVID-19 infection in liver transplant recipients. The authors achieved high positive humoral response by temporary suspend of anti-metabolite immunosuppressants such as mycophenolate mofetil and everolimus without increasing acute rejection in liver transplant recipients who take multiple (double or triple) immunosuppressants.
Minor comments
1. P2 Line57-62: In the background, the authors might better cite previously published papers (ex. No7) suggesting that steroids and mycophenolate mofetil are factors in low antibody titers of liver transplant recipients after vaccinations, as a reason for trying temporary suspension of immunosuppressants.
2. P3 Line107-113: How did the authors select patients who would be temporarily suspended immunosuppressants and those who would not?
P5 Line161: period before “patients” should be removed.
Author Response
Dear Reviewer:
We are pleased that we have the opportunity to revise our manuscript. We thank your comments and suggestions. Point-to-point responses are described as following:
- P2 Line57-62: In the background, the authors might better cite previously published papers (ex. No7) suggesting that steroids and mycophenolate mofetil are factors in low antibody titers of liver transplant recipients after vaccinations, as a reason for trying temporary suspension of immunosuppressants.
Reply: The Introduction section was revised and added “High dose steroids and mycophenolate mofetil (cellcept®, MMF) were reported as negative predictors of anti-covid-19 vaccine responses in liver transplantation”
- P3 Line107-113: How did the authors select patients who would be temporarily suspended immunosuppressants and those who would not?
Reply : The patients should have their liver transplantation for more than 6 months and have stable liver function for more than 3 months.
Thank you for your comments and suggestions again. Hopefully, our manuscript can be accepted.
Sincerely yours,
Wei-Chen Lee
Reviewer 3 Report
The authors present a study in which they determine optimum conditions for immunizing liver transplant patients with covid19 mRNA vaccine. The authors have compared outcomes by looking at a number of different variables related to the conditions of immunosupression associated with preventing transplant rejection. As such it is an interesting study. The authors point out one limitation which is that they are only looking at circulating antibody which may not be the correct endpoint. My only concern is with the actual determination of what is considered a positive response. The authors indicate in their materials and methods that they used what appears to be an arbitrary value to differentiate positive and negative responses to the vaccination. I think that the authors need to further discuss this point since it is critical to interpreting the results. Why did they use the value chosen? Is there evidence in the literature that this value represents an important number with respect to antibody protection against infection? This should not be hidden in the materials and methods but should be justified in the results and commented on in the discussion.
Author Response
Dear Reviewer:
We are pleased that we have the opportunity to revise our manuscript. We thank your comments and suggestions. Point-to-point responses are described as following:
Comment: My only concern is with the actual determination of what is considered a positive response. The authors indicate in their materials and methods that they used what appears to be an arbitrary value to differentiate positive and negative responses to the vaccination. I think that the authors need to further discuss this point since it is critical to interpreting the results. Why did they use the value chosen? Is there evidence in the literature that this value represents an important number with respect to antibody protection against infection? This should not be hidden in the materials and methods but should be justified in the results and commented on in the discussion.
Reply: The value to differentiate positive and negative responses to vaccination is not arbitrary. The antibody titers were determined by wild‐type SARS‐CoV‐2 neutralizing antibody assay performed at P3 laboratory. The adequate antibody production was referred to wild-type SARS-CoV-2 neutralization assay with 50% of cells free from infection, which was ≥ 9.62 IU/mL. [reference 10]
Thank you for reviewing our manuscript. We wish you can accept our explanation.
Best Regards,
Wei-Chen Lee
Reviewer 4 Report
Wei-Chen Lee et al. present in their manuscript „Adjustment of immunosuppressants to facilitate anti-Covid-19 antibody production after mRNA vacination in liver transplant recipients“ the results from a study assessing the immunogenicity of mRNA vaccination in liver transplant recipients who were treated with different immunosuppression regimens at the time of vaccination. Despite several limitations, the study provides important evidence on the anti-SARS-CoV-2 vaccination in specific populations.
Here, I have the number of comments and queries that should be reflected in the revised version of the manuscript:
1) Please, clearly state in the methods if the study was prospective or retrospective!
2) Please, specify the vaccination schedule used in your patients! The recommended interval between 2 doses of Moderna mRNA vaccine is 8 and 4 weeks in adults, and adolescents respectively. Why is the average interval 76-84 days in your patients?
3) Do you have any explanation for higher titers of neutralizing antibodies in the NA group compared to SS (Figure 1a)
4) Another limitation of the study is missing specific anti-SARS-CoV-2 antibody profiles including anti-RBD and anti-NCAP antibodies that may suggest previous contact with SARS-CoV-2. It is very important from the point of view of hybrid immunity (e.g. Kemlin et al., Hybrid immunity to SARS-CoV-2 in kidney transplant recipients and hemodialysis patients, American Journal of Transplantation). Please, discuss!
Thank you for the opportunity to review the manuscript!
Author Response
Dear Reviewer:
We are pleased that we have the opportunity to revise our manuscript. We thank your comments and suggestions. Point-to-point responses are described as following:
) Please, clearly state in the methods if the study was prospective or retrospective!
Reply: This study was prospective. We revise “All patients signed the consents and prospectively agreed to provide serum for antibody measurement”
2) Please, specify the vaccination schedule used in your patients! The recommended interval between 2 doses of Moderna mRNA vaccine is 8 and 4 weeks in adults, and adolescents respectively. Why is the average interval 76-84 days in your patients?
Reply: At that time, vaccines were lack in Taiwan. We could not give the patients vaccination as the schedule.
3) Do you have any explanation for higher titers of neutralizing antibodies in the NA group compared to SS (Figure 1a)
Reply: The median (interquartile) of antibody was 82.93 (22.63-229.37) IU/mL in SS group patients and 25.09 (9.62-455.27) IU/mL in NA group patients. The p value was 0.336. The titer in NA group was not higher than SS group.
4) Another limitation of the study is missing specific anti-SARS-CoV-2 antibody profiles including anti-RBD and anti-NCAP antibodies that may suggest previous contact with SARS-CoV-2. It is very important from the point of view of hybrid immunity (e.g. Kemlin et al., Hybrid immunity to SARS-CoV-2 in kidney transplant recipients and hemodialysis patients, American Journal of Transplantation). Please, discuss!
Reply: In Material section, we described that “All patients did not have the history of Covid-19 viral infection and SARS-CoV-2 serology test was not performed before anti-Covid-19 vaccination.”
In Discussion section, We add “We also did not measure specific anti-SARS-CoV-2 antibody profiles including anti-RBD and anti-NCAP antibodies to rule out previous contact with SARS-CoV-2 because there were only few people infected at that time in Taiwan.”
Thank you for reviewing our manuscript. We wish you accept our replies.
Best Regards,
Wei-Chen Lee
Round 2
Reviewer 3 Report
I believe that the authors have adequately addressed my concern. The manuscript is acceptable for publication.
Reviewer 4 Report
The authors reflected all proposals and responded to all queries appropriately. I recommend accepting the manuscript for publication.